# A Lightweight Visual Simultaneous Localization and Mapping Method with a High Precision in Dynamic Scenes

**DOI:** 10.3390/s23229274

**Published:** 2023-11-19

**Authors:** Qi Zhang, Wentao Yu, Weirong Liu, Hao Xu, Yuan He

**Affiliations:** 1School of Computer and Information Engineering, Central South University of Forestry and Technology, Changsha 410018, China; zhangqi@csuft.edu.cn (Q.Z.); xuhao@csuft.edu.cn (H.X.); heyuan@csuft.edu.cn (Y.H.); 2School of Computer, Central South University, Changsha 410083, China; frat@csu.edu.cn

**Keywords:** VSLAM, dynamic environment, target detection, lightweight network, communication consistency, attention mechanism

## Abstract

Currently, in most traditional VSLAM (visual SLAM) systems, static assumptions result in a low accuracy in dynamic environments, or result in a new and higher level of accuracy but at the cost of sacrificing the real–time property. In highly dynamic scenes, balancing a high accuracy and a low computational cost has become a pivotal requirement for VSLAM systems. This paper proposes a new VSLAM system, balancing the competitive demands between positioning accuracy and computational complexity and thereby further improving the overall system properties. From the perspective of accuracy, the system applies an improved lightweight target detection network to quickly detect dynamic feature points while extracting feature points at the front end of the system, and only feature points of static targets are applied for frame matching. Meanwhile, the attention mechanism is integrated into the target detection network to continuously and accurately capture dynamic factors to cope with more complex dynamic environments. From the perspective of computational expense, the lightweight network Ghostnet module is applied as the backbone network of the target detection network YOLOv5s, significantly reducing the number of model parameters and improving the overall inference speed of the algorithm. Experimental results on the TUM dynamic dataset indicate that in contrast with the ORB–SLAM3 system, the pose estimation accuracy of the system improved by 84.04%. In contrast with dynamic SLAM systems such as DS–SLAM and DVO SLAM, the system has a significantly improved positioning accuracy. In contrast with other VSLAM algorithms based on deep learning, the system has superior real–time properties while maintaining a similar accuracy index.

## 1. Introduction

Simultaneous Localization and Mapping (SLAM) refers to the process in which a robot perceives the environment and estimates its own state in an unidentified environment through its own sensors. Common sensors carried by robots include cameras, laser sensors, inertial sensors, etc. As a result of the broader range of colors and texture information in the images captured by cameras, as well as their low price, small size, and low power consumption, they are extensively used. A system in which the camera serves as the primary sensor is called a visual SLAM system [1]. In recent years, visual SLAM has been popularly applied in many application fields such as virtual reality (VR) [2], augmented reality (AR) [3], unmanned aerial vehicle (UAV) [4] or unmanned ground vehicle (UGV) navigation [5], autonomous mobile robots [6], and so on.

A high precision and a low computational cost are the two pivotal requirements for VSLAM [7]. In recent years, many solutions have been proposed for VSLAM, such as dense tracking and mapping (DTAM) [8], large–scale direct monocular SLAM (LSD–SLAM) [9], semidirect visual odometry (SVO) [10], and ORB–SLAM2 [11]. Traditional vision–based SLAM research has demonstrated many achievements, but it may not achieve the desired results in challenging environments. The above SLAM schemes are all based upon the assumption of static scenes, but the existence of dynamic objects in real scenes is ineluctable. If dynamic objects have strong texture information, the system will extract multitudinous features from the dynamic objects. Tracking unstable feature points will profoundly affect pose estimation, causing significant trajectory errors and even tracking losses, making it laborious to ensure the accuracy of the entire SLAM system.

For the sake of improving the accuracy of VSLAM systems in dynamic scenes, some algorithms eliminate the impact of dynamic objects through the movement law of the camera or geometric model. However, the majority of algorithms have rigorous restrictions or a low model accuracy which restrict accuracy improvements. In recent years, with the development of deep learning, many scholars have applied semantic segmentation networks to identify and remove dynamic objects from images. Although semantic segmentation networks can effectively partition dynamic objects, the large architecture of most semantic segmentation network models enormously increases the system’s computational expense and cannot meet real–time requirements. Thus, it is notably important to accurately eliminate dynamic factors in the environment while ensuring the real–time property of the VSLAM system.

This paper proposes a new, high–precision, lightweight VSLAM system to address the above issues. At the fore–end of the VSLAM system, an improved lightweight target detection network is applied to quickly detect dynamic feature points while extracting ORB feature points. Subsequently, a logical discrimination module is applied to eliminate dynamic feature points. Meanwhile, attention mechanisms are combined in the target detection network to continuously and accurately capture dynamic factors for the sake of coping with more complex dynamic environments. With the assistance of communication consistency protocols, the target detection network and logical discrimination algorithms work together to enhance the real–time property and robustness of the system in dynamic environments. The primary contributions of this paper are as follows:This article proposes an efficient and lightweight visual SLAM scheme based on ORB–SLAM 2 by optimizing the fore–end of traditional ORB–SLAM 2 systems. This scheme can balance the competing requirements between positioning accuracy and computational complexity, further improving the overall system performance and enabling it to cope with challenging complex dynamic environments;For the sake of improving the accuracy and real–time property of the system, we have designed a new network architecture to diminish the parameters and mapping redundancy of the target detection network. In the meantime, an attention mechanism module is integrated to better capture high–dynamic objects in spatial backgrounds. The logical discrimination algorithm is combined with the ORB–SLAM2 system to efficiently remove dynamic feature points. The system adopts the conformance of the communication protocol to efficiently integrate the preprocessing part in parallel in the form of threads, achieving an efficient and lightweight system;The public TUM dataset was used for qualitative and quantitative evaluations. In contrast with traditional VSLAM systems, the system accuracy was significantly improved. In contrast with other deep–learning–based VSLAM algorithms, the system has a better real–time performance while maintaining a similar accuracy index.

## 2. Related Works

With regard to the processing of moving objects, most classic SLAM systems treat them as noise and then eliminate the noise through the random sample consensus (RANSAC) [12] or the robust cost function [13]; however, this method will cease to be effective when there are massive dynamic objects in the scene. Tan et al. [14] improved the robustness of the traditional RANSAC scheme by changing it to an adaptive one, as well as adding color information and visual geometric constraints, but the accuracy improvement is still finite. Zou et al. [15] projected the feature points from the previous frame onto the current frame and calculated their re–projection error. When the error exceeds a certain threshold, the feature point will be eliminatedd as a dynamic feature point. Li et al. [16] adopted the concept of deep edge points by estimating the probability that edges belong to static points in consecutive image frames, and added this probability as a weight to the pose matching process. This not only diminishes the weight of the optimization term where dynamic points are located, but also to some extent diminishes the impact of dynamic points. However, it does not achieve overall exclusion of the influence of dynamic points. Kim et al. [17] took advantage of the spatial continuity of feature points on the same object to differentiate range images and obtain dynamic objects in the image. However, this method only focuses on depth variation and cannot cover all dynamic regions. Liang et al. [18] took advantage of the spatio–temporal continuity of feature points between images to improve the above scheme and achieved superior results, but still did not achieve competitive actual performance. Sun et al. [19] came up with the fundamental matrix by matching the color information of feature points between two frames, estimated dynamic feature points based on this, and then applied the particle filtering method to track these feature points, achieving a dynamic and static segmentation of feature points. Moratuwage et al. [20] proposed a random finite set based oupon feature maps and actual measured values to track feature points, and obtained estimates of dynamic features through Bayesian recursive estimation of probability density. Chivilo et al. [21], Handa et al. [22] applied changes in pixel optical flow values to determine dynamic feature points in an image. Liu et al. [23] applied dense optical flow method to predict semantic labels in images and obtain dynamic semantic information. These algorithms have superb performance in static environments. Dai et al. [24] applied the triangulation method to connect scene map points into triangles, and compared the changes in the triangle edges in the map points. The edges with significant changes were eliminated, and the area of the remaining connected triangles in various regions was calculated. By assuming that the largest area of the connected region was a static region, various dynamic objects in the scene could be filtered out with a bang, but the effect of eliminating dynamic factors was unsatisfactory. Yang et al. [25] also adopted a triangulation edge scheme, only using different methods to determine dynamic points for feature points in the image, which also achieved preferable results. However, there is still a problem of insufficient accuracy. The above algorithms all remove the detected dynamic feature points to improve the accuracy of the system in dynamic environments. But most of the above algorithms cannot accurately distinguish between dynamic and static feature points by reason of strict constraints or low mathematical model accuracy, resulting in insufficient system accuracy.

With the development of deep learning algorithms and the improvement of computer performance, significant progress has been made in target detection and semantic segmentation methods [26], and researchers are gradually realizing that these methods may be conducive to solve the SLAM problem mentioned above. The combination of traditional VSLAM technology and deep learning based semantic segmentation and target detection methods can significantly improve the robustness and accuracy of SLAM systems in dynamic environments. Bescos et al. proposed DynaSLAM [27], which makes use of Mask R–CNN(Regions with CNN features) [28] to obtain semantic segmentation results and determine possible moving feature points. In the meantime, the method of multi–view geometry is applied to detect dynamic objects in the image that have not been detected by semantic segmentation, and then the two detection results are incorporated. For a feature point, as long as one of the two detection results is dynamic, it is considered dynamic and deleted. Simultaneously, this scheme also applies camera motion information to repair the segmented area of the dynamic target from different perspectives, thereby the system is able to obtain a complete map. However, owing to the use of Mask R–CNN network and background repair function, the system is laborious to run in real–time. Yu et al. [29] added a semantic segmentation thread to DS–SLAM based on ORB–SLAM, which applies a Segnet network to process images and then transmits them to the primary thread. The thread combines the optical flow constraint scheme in the tracking thread to eliminate dynamic objects. However, the process of extracting dynamic information using semantic segmentation is time–consuming and cannot meet the real–time requirements of the system. Yang et al. [30] proposed SGC–VSLAM, which extracts semantic detection frames through YOLO and eliminates dynamic features with semantic and geometric constraints. Long et al. [31] combined the multiple view geometry of the optimal error compensation homography matrix of the semantic segmentation network PSPNet [32] to eliminate dynamic feature points, and added a search strategy of reverse ant colony to the multiple view geometry, improving the real–time property of the system. Zhang et al. [33] applied semantic information to estimate rigid objects in the scene. Yuan et al. [34] constructed a bag–of–words model for semantic tags to diminish the impact of dynamic objects on SLAM systems. Although the above algorithms have overall improved the accuracy of VSLAM, the integration of multitudinous semantic information bring about the system not meeting real–time requirements. In the context of practical industrial applications, real–time property is one of the important criteria for determining whether a VSLAM system is outstanding.

Most of the research on VSLAM systems in dynamic environments has focused on improving the positioning accuracy of robots in terms of visual odometer positioning accuracy. However, the proposed model is hard–pressed to meet the real–time requirements of mobile robots, and there is relatively little research on how to more accurately exclude the influence of dynamic factors in complex and changing environments. The optimization of VSLAM in dynamic scenes can be approached from two aspects: one is the accuracy of camera pose estimation, and the other is the overall real–time property of the system [35]. In recent years, most scholars have been inspired by deep learning and integrated it into VSLAM systems. Although the VSLAM system combined with target detection network ensures real–time property, there is still room for improvement in system accuracy [36]. Integrating semantic segmentation networks into the VSLAM framework to extract dynamic information from images, while greatly ensuring the accuracy of the system, sacrifices the real–time property of the system owing to the large scale of the network model [37]. How to balance accuracy and real–time property has become the primary issue for VSLAM systems. In response to this issue, this paper proposes a VSLAM system pose optimization method based on an improved lightweight target detection network, on account of the research of numerous outstanding scholars. The system is based on the ORB–SLAM2 architecture and applies an improved target detection network to quickly detect all recognizable objects in the current frame, obtain object semantic information and bounding boxes, distinguish potential moving objects based on semantic information, and combine attention mechanism with the target detection network to continuously and accurately capture dynamic factors to cope with more complex dynamic environments, further improving the positioning accuracy of the system in complex dynamic environments. This system can extract ORB feature points in the fore–end of the SLAM system while quickly and effectively eliminating dynamic feature points. By only taking advantage of feature points on static targets for frame matching, the positioning accuracy of the system in dynamic environments is improved. At the same time, the competition between positioning accuracy and computational complexity is further balanced through design, taking into account the high precision and low computational cost requirements of the VSLAM system, thereby improving the overall performance of VSLAM. The experimental results indicate that the scheme proposed in this paper can effectively improve the performance of VSLAM in high dynamic scenes.

## 3. System Overview

The VSLAM scheme proposed in this paper mainly focuses on preprocessing and optimization of the system. On the basis of the classic open–source system ORB–SLAM2, the improved target detection module is embedded in the VSLAM system in parallel in the form of threads. An RGBD camera is adopted which can not only obtain color images of the scene, but also obtain the depth value corresponding to each pixel in the color image. Firstly, an ORB feature extraction module is applied to extract the feature points of the current frame. In the meantime, an improved target detection module is introduced to detect the semantic information of the scene and obtain the semantic boundaries of the objects. In the light of the semantic information, a logical discrimination mechanism is used to distinguish potential dynamic objects from static objects; Subsequently, determine whether to eliminate the feature point on the basis of whether its coordinates fall in the dynamic or static region; Finally, the remaining static feature points are applied for feature matching to calculate camera pose, thereby optimizing one’s own position while auditing and generating keyframes, laying the foundation for subsequent LocalMapping and LoopClosing threads.

### 3.1. SLAM System Framework

The SLAM system in this paper is ameliorated on the basis of ORB–SLAM2 system framework. The RGB–D camera is adopted to obtain scene images. ORB–SLAM2 runs in parallel on the basis of the thread framework, including target detection, tracking, local map, local loopback and map construction. The SLAM system framework in this paper is shown in Figure 1. Firstly, the collected images are processed by an optimized target detection network to extract dynamic object information in the scene. The tracking thread is applied to extract ORB feature points from the image. After removing the feature points inside the dynamic target detection frame mask, the remaining feature points are applied for feature matching to restore the camera pose. After obtaining the initial pose, local pose optimization is carried out, followed by the review and generation of keyframes. When the queue of keyframes to be processed is not empty, the LocalMapping thread starts working. In this thread, the keyframes are processed, deleting MapPoints that do not meet the conditions and creating new MapPoints with matching relationships for supplementation. Then, local BA optimization is carried out to optimize all MapPoints and delete redundant keyframes, and add the current keyframe to the closed–loop detection. The loop detection thread is applied to detect large loops and correct accumulated errors by performing pose optimization. After optimizing the pose map, this thread will start a fourth thread to execute the global BA and calculate the optimal structure and motion results.

To improve the accuracy and robustness of the ORB–SLAM system in dynamic environments, this paper adds a target detection module as the fourth thread on top of the original three parallel threads of the system. In traditional tracking threads, if the extracted ORB feature points remain in motion, the error of localization will continuously accumulate in the process of feature point matching and pose calculation, ultimately giving rise to localization failure. As shown in Figure 2, a target detection thread is added to the tracking thread to detect and eliminate dynamic ORB feature points on dynamic targets, as a result they do not participate in camera pose calculation, and only static feature points remain to participate in pose calculation.

### 3.2. Object Detection Network Based on Lightweight Yolov5s

The system in this paper applies YOLOv5s network for dynamic object detection. YOLOv5 target detection network is an efficient and powerful target detection model, which has high accuracy in real–time target detection algorithms and achieves the optimum balance between accuracy and speed. The YOLOv5 target detection network structure applies CSPMarknet53 as the primary network, retains the head part of YOLOv3, and applies the spatial pyramid pooling idea to increase the receptive field, which can separate the most important context features without decreasing the speed of network operation. Simultaneously, the path polymerization module in PANet is applied to change the fusion method from addition to multiplication, significantly improving the efficiency of the module. The system in this paper applies the YOLOv5s network to efficiently detect dynamic objects in dynamic scenes. If the results of target detection contain dynamic objects, a mask region corresponding to the bounding box is generated. Combining the mask region and the results of logical discrimination, the dynamic ORB feature points in the mask region are eliminated, which can significantly diminish the impact of dynamic objects on the VSLAM system.

As a regression–based target detection algorithm, it solves the problem of target detection as a regression problem, directly obtaining the predicted bounding box position of an object and classification from an input image frame. It ensures accuracy while also making allowances for real–time property, achieving a superb balance between speed and accuracy. YOLOv5 is currently undergoing iterative updates, including 4 versions: YOLOv5s, YOLOv5m, YOLOv5l, and YOLO5x. The principal difference between various versions lies in the depth and width of feature maps. The YOLOv5s network is the network with the smallest depth and feature map width in the YOLOv5 series. The model parameter quantity is about 7.5 M, and on V100GPU, the inference speed for each image is only 0.002 s, which means the frame rate is 500 frames/s, fundamentally meeting the real–time detection requirements of the VSLAM system in this paper. YOLOv5s consists of three parts, namely backbone, neck, and head. Backbone: This is a feature extraction network that extracts information from the image for the following network. Neck: The neck is between the backbone and head, and further employs the features extracted via the backbone to improve the robustness of the model. Head: This takes the network output and makes a prediction using the previously extracted features. The yolov5s network structure is shown in Figure 3 below.

#### Lightweight Network Ghostnet

As shown in Figure 4, the convolutional neural network has a large demand for internal memory and a large amount of computation, which makes it unable to run in real time on mobile devices, and also limits its real–time property on the VSLAM system. In preeminent CNN models, the redundancy of feature maps is extremely noteworthy, but few people consider the redundancy in feature map problems in model structure design, considering that the redundant information in these feature map layers may be a significant component of a successful model. It is precisely because these redundant information ensure a comprehensive understanding of the input data that this lightweight model does not attempt to remove these redundant feature maps, but instead attempts to obtain these redundant feature maps using lower cost computational complexity. Finally, the first part is treated as an identity map and concated with the results of the second step, as shown in Figure 4. Plug and play: The Ghost Module is a plug and play module that seamlessly connects to existing CNN. The Ghost bottlenecks, composed of Ghost Modules, were designed with Ghost Net [38]. On ILSVRC–2012, the top 1 exceeded Mobilenet–V3 [39] and had fewer parameters.

In VSLAM methods that combine deep learning, real–time property is even more important. This paper first combines pixel based semantic segmentation methods with SLAM systems, but experiments have found that pixel based semantic segmentation methods are laborious to run in real–time. As a result, this paper introduces a deep learning based target detection network to detect potential moving objects in the environment, and further ameliorates it by integrating lightweight networks. This method meets the real–time requirements of SLAM systems with high running speed.

The YOLOv5 backbone feature extraction network adopts a CSP(Communicating Sequential Processes) structure, which brings multitudinous parameters and slow detection speed, making it laborious to be applied in certain real application scenarios such as mobile or embedded devices. Firstly, the model is too large and faces the problem of inadequate memory. Secondly, these scenarios require low latency or fast response time. Thus, studying small and efficient CNN models is vital in these scenarios. In response to the detection difficulties in specific application scenarios such as VSLAM systems, a new network structure is proposed to ensure the inference speed and accuracy at the same time.The improved backbone network is shown in Figure 5. Firstly, the current input frame is processed into an image input with a size of 640 × 640 × 3, and then the model layer factor is defined as 0.33 to adjust the depth of the network, and the model channel factor is defined as 0.50 to adjust the depth of the network. Firstly, the image is sent to the backbone network for processing, and it is firstly processed by Conv convolution. The size of the feature map becomes 1/2 of the original image, and then after Ghostconv lightweight convolution operation, the size of the feature map becomes 1/4 of the original image, and then after C3 structure processing, the input is processed in two ways. One is to obtain an output value output1 from the residual network, the other is to obtain an output matrix output2 directly from the Conv convolutional network, and finally add the two output values. The size of the feature map does not change through the C3 layer, and then the size of the feature map becomes 1/32 of the original image after three layers of Ghostconv processing. Then through the Coordinate Attention layer, this module is used to enhance the feature correlation between different channels. Finally, through the SPPF (Spatial Pyramid Pooling) layer, which is a spatial pyramid pooling network structure, the layer is used to extract the receptive field features of different sizes to cover different scales of the target. This structure can make the network better adapt to objects of different scales, improve the accuracy of object detection, and ensure the rapidity of detection.

## 4. Detection of Moving Objects and Elimination of Dynamic ORB Feature Points

The current prevalent SLAM system’s visual odometer generally assumes that map points in the environment are fixed and unchanging. As a result, the pixel value difference between the two camera images of the same map point in the environment should only result from the camera’s pose change, so that the camera’s pose change can be obtained. In Figure 6, the impact of dynamic objects in the scene on pose estimation of visual odometer is depicted from the perspective of camera observation.

At the *t* − 1 moment, the camera obtains the initial position *P_t_*_−1_ by observing the static feature *s* and the dynamic feature *D_t_*_−1_. At the *t* moment, traditional visual odometers assume that the dynamic features are not moving and are still at the *D_t_*_−1_ position, meaning that only the camera is moving, resulting in the camera moving to the $P_{t}^{\prime }$ position. However, in fact, the camera position *P_t_* should be obtained by the motion feature *D_t_*. For the same reason, the estimated position $P_{t+1}^{\prime }$ of *t* + 1 does not match the actual position, and the actual position *P_t_*_+1_ of the camera is shown in the figure. Connect the true position of the camera with a solid line, that is, the solid line with an arrow in the figure represents the real motion trajectory of the camera. Connect the estimated position of the camera with a dotted line, that is, the dotted line with an arrow in the figure represents the estimated trajectory of the camera. The discrepancy between the two trajectories indicates that dynamic objects in the environment have a significant impact on the visual odometer, resulting in incorrect camera pose estimation. Therefore, handling dynamic objects in the environment is deadly serious for visual odometers.

Common feature points include SIFT, SURF, ORB, etc. Among them, SIFT feature points fully consider the changes in lighting, scale, rotation, etc. that occur during image transformation, but with it comes a huge computational burden. SURF feature points also face the problem of high computational burden. By reason of the fact that the extraction and matching of image features throughout the entire SLAM process is only one of many steps, real–time calculation of SIFT and SURF features for localization and mapping still faces challenges. Therefore, this “extravagant” image feature is rarely applied in SLAM. As a result, a more robust SLAM system should follow with interest real–time issues, extracting 1000 feature points in the same image simultaneously. The SIFT cost is about 5228.7 ms, SURF cost is about 217.3 ms, and ORB cost is about 15.3 ms. From this, it can be seen that the ORB feature points have significantly improved in speed, which better meets the requirements of VSLAM systems with high real–time requirements. Contrast to the sparse optical flow method with insufficient accuracy and the dense optical flow method with high computational complexity and poor real–time property [40], the logical discriminant method differentiates from the optical flow method, which has a more stringent premise. It can better meet the high demand for real–time property of VSLAM systems in complex dynamic scenes, thereby improving the overall performance of the system.

### 4.1. Detection and Elimination of Dynamic Feature Points

The improved lightweight target detection network in this paper generates a prediction box based on image inference, consisting of six parameters: positional parameters *x*, *y*, *w*, *h*, as well as confidence and classification results. *x*, *y* represents the relative values of the center of the prediction box and the original image, while *w* and *h* represent the relative values of the length and width of the prediction box and the original image. Confidence represents the credibility of the object contained within the prediction box and the accuracy of the prediction box’s position. The classification results are determined based on the object categories contained in the dataset used during training. As shown in Figure 7, taking the upper left prediction box as an example, this paper first converts its positional parameters *x*, *y*, *w*, and *h* into the coordinates of the upper left and lower right vertices of the prediction box in the original image, set to (*X_A_*_1_, *Y_A_*_1_), and (*X_A_*_2_, *Y_A_*_2_) respectively (in the original image, the upper left vertex is the coordinate origin, set to the right is the positive *x*–axis direction, and set to the downward direction is the positive *y*–axis direction).

Next, the parameters output by the target detection network were converted into a formula for the coordinates of the target detection box in the original image, as shown in Formula (1):(1)$$\{\begin{array}{l} \left(X_{A1}+X_{A2}\right)/2/l=x\\ \left(Y_{A1}+Y_{A2}\right)/2/d=y\\ \left(X_{A2}-X_{A1}\right)/l=w\\ \left(Y_{A2}-Y_{A1}\right)/d=h \end{array}$$
where *l* is the width of the original image; *d* is the height of the original image. The ORB feature points contained in the prediction box are all undetermined dynamic feature points based on transcendent knowledge judgment, and the static feature points outside the prediction box. Firstly, assume that all feature point sets are *P* = {*P*_1_, *P*_2_, …, *P_n_*}, the set of undetermined dynamic feature points is *R*_1_ = {*R*_1_, *R*_2_, …, *R_n_*}, and the set of static feature points is *O* = {*O*_1_, *O*_2_, …, *O_n_*}, *P* = *R*∪*O*. All feature points in the set will participate in the screening of dynamic feature points, and the coordinate information of each feature point (*X*, *Y*) will be calculated by the ORB–SLAM2 system fore–end.

Meanwhile, in this paper, the communication method of UNIX domain sockets is applied to achieve real–time communication of VSLAM systems. The various communication protocols in the socket module can achieve various communications, including TCP/IP. In this paper, the UNIX domain protocol is applied to omit the process of storing RGBD images, accelerate system operation speed, and simultaneously transmit one frame of preprocessed image data to the VSLAM system, solving the problem of asynchronous and non real–time throughout the previous process. The lightweight target detection network is truly integrated with the SLAM system in the form of parallel threads, and the dynamic feature points of key frames are efficiently and real–time detected, preparing for the subsequent elimination of dynamic feature points. The dynamic feature point elimination algorithm in this paper is shown in Algorithm 1:
**Algorithm 1** Dynamic Feature Point Logical Discrimination Mechanism.**Input:** Current keyframe **I_r_** (Contains a variety of features); A keypoint on the reference frame *I_r_*(x,y);1: Classifies objects into three states: highly dynamic, medium dynamic, and low dynamic (directly considered static);2: **if**
*I_r_*(x,y) is in the low dynamic feature category **then** it is determined to be a static feature point;}  **else**
*I_r_*(x,y) is in the high dynamic feature range  {    **if**
*I_r_*(x,y) is in the low dynamic feature category **then** it is determined to be a static feature point;   }    **else** *I_r_*(x,y) is determined to be a dynamic feature point 3: Dynamic feature points will be eliminated, static feature points will be used for subsequent inter–frame matching4: **Return** Keyframes processed Ir∗**Output:** Strips the current keyframe of dynamic feature points

### 4.2. The Perceptual Function of CA Attention Mechanism in Dynamic Environments

In the VSLAM system, the movement of targets in a dynamic environment undergoes spatial changes, either by maintaining equidistant motion along parallel trajectories in front of the camera, or by moving perpendicular trajectories near or far from the camera. Accompanying it is the varying magnitude of dynamic factors in the image, which can affect the recognition accuracy and decrease the removal accuracy when the dynamic target becomes too small. For the sake of better identifying and eliminate dynamic factors, the SLAM system in this paper introduces an attention mechanism in the fore–end image preprocessing module of the system. The lightweight target detection network is applied as the carrier to integrate the CA (coordinate attention for effective mobile network design) attention mechanism [41], in an effort to improve the VSLAM system’s attention to dynamic factors and improve recognition accuracy. Contrast to the computational overhead of most attention mechanisms, CA attention mechanism is more suitable for mobile networks with limited computing power. In the meantime, coordinated attention encodes channel relationships and long–term dependencies through precise positional information, compensating for the significance of other attention only considering encoding channel information and neglecting positional information. This is crucial for capturing dynamic object structures in SLAM visual tasks.

A coordinate attention module can be seen as a computing unit intended to enhancing the expression ability of features in Mobile Network. It can take any intermediate feature tensor $X=[x_{1},x_{2},\ldots ,x_{C}]\in R^{C\times H\times W}$ as input and output a $Y=[y_{1},y_{2},\ldots ,y_{C}]$ with the same size as the tensor and enhanced representation through transformation. Coordinated attention encodes channel relationships and long–term dependencies through precise positional information. The specific operation is divided into two steps: Coordinated information embedding and coordinated attention generation. The coordinated attention module structure is shown in Figure 8.

The global pooling method is commonly applied for the global encoding of channel attention-encoded spatial information, but by reason of its compression of global spatial information into channel descriptors, it is laborious to save positional information. In an effort to enable the attention module to capture remote spatial interactions with precise positional information, this paper decomposes global pooling into a pair of one–dimensional feature encoding operations in conformity to the following formula:(2)$$z_{c}=\frac{1}{H\times W}\sum_{i=1}^{H}\sum_{j=1}^{W}x_{c}(i,j)$$

Specifically, given the input *X*, each channel is first encoded along the horizontal and vertical coordinates using a pooling kernel with dimensions (*H*, 1) or (1, *W*), respectively. As a result, the output of channel *c* with a height of *h* can be expressed as:(3)$$z_{c}^{h}(h)=\frac{1}{W}\sum_{0\leq i\leq W}^{H}x_{c}(h,i)$$

In the same way, the output of channel *c* with a width of *w* can be written as:(4)$$z_{c}^{w}(w)=\frac{1}{H}\sum_{0\leq i<W}x_{c}(j,w)$$

After the transformation in information embedding, this part concatenates the above transformation and then applies convolutional transformation functions to transform it:(5)$$f=\delta (F_{1}([z^{h},z^{w}]))$$
(6)$$g^{h}=\sigma (F_{h}(f^{h}))$$
(7)$$g^{w}=\sigma (F_{w}(f^{w}))$$

In the last resort, the output Y of the Coordinate Attention Block can be written as:(8)$$y_{c}(i,j)=x_{c}(i,j)\times g_{c}^{h}(i)\times g_{c}^{w}(j)$$

The involvement of the CA attention mechanism has improved the performance of lightweight networks in three ways. Firstly, for applications in the Mobile environment, the new transformation should be as simple as possible; Secondly, it can fully utilize the captured location information to accurately capture regions of interest; Finally, it should also be able to effectively capture the relationships between channels. As shown in Figure 9, we can see that in the actual dynamic environment, the dynamic character on the right continuously moves from point A to point C and changes in position in space:

When dynamic targets constantly change their spatial position in a mobile environment, the proportion of dynamic targets in the RGBD image frame input to the SLAM system will continuously change, which will affect the detection of dynamic targets by lightweight networks, and subsequently affect the next step of removal work, ultimately affecting the overall performance of the system. So when we integrate the attention mechanism and take high moving objects as our object of interest, they will gain higher attention and be more comfortably perceived and captured by the network, providing superior results for eliminating dynamic features. The dynamic capture effect with higher attention is shown in Figure 10:

Figure 11 shows the comparison of the estimated trajectories of the two systems on the walking_xyz dataset (attention mechanism module added on the left). The left estimated trajectory results are based on the Yolov5s_ghostnet_Ca network model, and the right estimated trajectory results are based on the Yolov5s_ghostnet network model (the training environment of the two network models is the same).

(1)The gray dotted line represents the true trajectory of this dataset, which is the groundtruth (reference); the colored dashed lines represent the system estimation results;(2)Compared with the left, the estimated trajectory results of the system without attention mechanism on the right show excessive overlap between the estimated trajectory and the true value at the tip, and the direction of the colored dotted line at the tip is chaotic and not clear (the end of the trajectory is zoomed in as shown in the left image of Figure 12), so the performance of the system can be preliminarily judged;

As shown in Figure 12, we can see it in detail that the VSLAM system incorporating CA attention mechanism performs preferably in the four accuracy error evaluation criteria of Max, Min, RMSE, and Std. It can be seen that the integration of CA attention mechanism can effectively help the VSLAM system decrease errors and increase accuracy.

**Figure 12 sensors-23-09274-f012:**
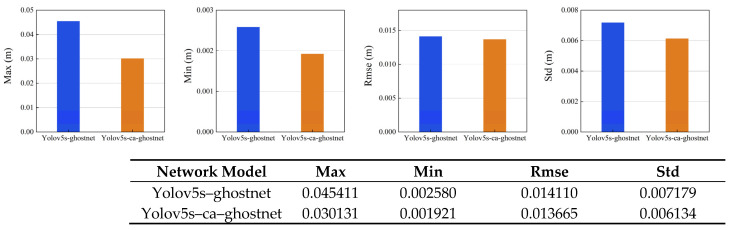
Comparison between the Two Systems in Terms of Maximum Error, Minimum Error, Root Mean Squared Error and Standard Deviation (On the right is the attention mechanism module added). The specific data are shown in the table.

## 5. Experiments and Results Analysis

This paper applies three data sets in the RGB–D data set series provided by Technical University of Munich for testing, namely walking_static, walking_xyz, walking_halfsphere. This series is divided into high dynamic scenes and low dynamic scenes. In high dynamic scenes, people will continue to walk in the scene; In low dynamic scenarios, people do not have obvious movements. The static, xyz, and halfsphere in the dataset name represent various camera motion modes: the camera is basically stationary; The camera moves along the *x* and *y* axes; The camera rotates on the surface shift, pitch, and yaw axes of a hemisphere with a diameter of 1 m. The real trajectory in the dataset is obtained by a motion capture system that includes multiple high–speed cameras and inertial measurement systems, which can obtain real–time data on camera position and attitude. Thus, this dataset has been adopted by most VSLAM researchers as one of the standard datasets for evaluating VSLAM systems.

### 5.1. Dynamic Target Detection Experiment

The target detection network is trained applying the COCO dataset, which is a large–scale dataset that can be applied for image detection, semantic segmentation, and image captioning. It has over 330 K images (220 K of which are labeled images), including 1.5 million targets, 80 target categories (object categories: pedestrians, cars, elephants, etc.), 91 material categories (stuff categories: grass, walls, sky, etc.), each image containing five sentence descriptions of the image, and 250,000 pedestrians with key point annotations. This data set is enough to meet the detection of dynamic targets, so as to better detect the dynamic factors in the SLAM environment and eliminate them in the next step. The training sample label and confusion matrix are shown in Figure 13. The loss of the training process until the prediction box, category, and confidence is steadily decreasing. For the sake of achieving better training effects, the batch size is then reduced and trained until the model converges. The accuracy and confidence and recall and confidence training curves of the proposed model are shown in Figure 14.

### 5.2. Pose Estimation Error Analysis Experiment

Experiment on error analysis of pose estimation applies the evo tool to test and compare the camera pose cameratrajectory.txt estimated by the ORB–SLAM system with the real pose groundtruth. txt given by the dataset. The test indicator is absolute trajectory error (ATE), which directly calculates the difference between the true camera pose value and the estimated value of the SLAM system, which can intuitively reflect the algorithm accuracy and global consistency of the trajectory. In the experiment, the root mean squared deviation (RMSE) is applied as the main evaluation standard of system performance. RMSE is applied to describe the deviation between the observed value and the true value, which is vulnerable to large or accidental errors, so it can better reflect the robustness of the system. The ATE definition of frame is as follows:(9)$$F_{i}:=Q_{i}^{-1}SP_{i}$$
where *P*_1_, …, *P_n_*
∈ *SE* (3) is used to estimate the pose, *Q*_1_, …, *Q_n_* ∈ *SE* (3) is the real pose. It should be noted that the estimated pose and groundtruth are usually not in the same coordinate system. As a result, we need to align the two. For binocular SLAM and RGB D SLAM with uniform scales, we need to calculate a transformation matrix (3) from the estimated pose to the real pose using the least squares method. Then the RMSE is expressed as follows:(10)$$\mathrm{RMSE}\left(F_{1:n},\Delta \right):={\left(\frac{1}{m}\sum_{i=1}^{m}{\left\|\mathrm{trans}\left(F_{i}\right)\right\|}^{2}\right)}^{\frac{1}{2}}$$
where, (1…, *n*) represents time (or frame). Here, we assume that the estimated pose and the real pose are aligned at each frame time, and the total number of frames is identical. Δ represents the time interval. Given the total number of frames *n* and interval Δ, *m* = *n* – Δ. ATEs can be obtained, where trans (*F_i_*) represents translation error.

The following figures show the system’s performance in walking_xyz, walking_halfsphere. The estimated trajectories and error distribution under two datasets are shown in Figure 15 and Figure 16:

The system in this paper is contrast to different SLAM systems for error, and the error ATE is statistically compared, as shown in Table 1, the data and information visualization comparison is shown in Figure 17:

The VSLAM system designed in this paper is contrast with SLAM systems in other dynamic scenes, including DynaSLAM and DS–SLAM systems based on semantic segmentation networks, DVO–SLAM that obtains constraints by optimizing the pose map calculation between key frames to minimize photometric and depth errors, while OFD–SLAM and MR–SLAM dynamic SLAM methods based on the optical flow. The comparison results are shown in Table 2. Among them, “—” indicates that this dataset was not tested in the original paper. From the Table 2, it can be found that the proposed system effectively maintains a minimum error accuracy on the first two data sets. On the third dataset the errors of the system in this paper and DynaSLAM remain at the same level.The root mean squared deviation error (RMSE) of this system under walking_static, walking_xyz and walking_halfsphere data is comprehensively compared. The error analysis is shown in Figure 18 below:

If the error of ORB–SLAM3 is *m* and the error of the system in this paper is *n*, the calculation formula for the relative improvement rate *R* is:(11)$$R=\frac{m-n}{m}\times 100\%$$

From Table 3, it can be seen that contrast with the ORB–SLAM3 [42] system, the average improvement rate of RMSE in this system under three datasets is 84.04%. Regarding the issue that the enhancement effect in the walking_static dataset is not as good as the other two datasets, by observing the images processed by the target detection network and the actual state of the system during operation, it was found that in the walking_static dataset, a considerable portion of image frames have characters occupying too much of the frame area, resulting in fewer detected static feature points, and there is image distortion caused by excessive image rotation angle in the dataset, Some portraits were not detected by improved target detection networks, ultimately resulting in a decrease in estimation accuracy.

### 5.3. Efficiency Test of the Proposed Algorithm

The number of model parameters, computational power requirements, inference time, and frame rate of different object detection networks on GPU and CPU are shown in Table 4 and Table 5 for detection inference on 826 images in one of the datasets. All tests were performed five times and the final results were averaged.

In order to highlight the real–time performance of the algorithm in this paper, we compare with several mainstream visual SLAM algorithms based on deep learning, as shown in Table 6. It can be seen from the table below. DynaSLAM, DM–SLAM, and Detect–SLAM take a long time to process each frame due to the semantic thread, so the tracking thread takes more than 200 ms to process each frame. RDS–SLAM and DS–SLAM with excellent real–time performance also take more than 50 ms to process each picture frame in the tracking thread. Compared with other visual SLAM algorithms based on deep learning, the tracking thread of the proposed algorithm only takes 44.77 ms to process each frame, which is better than other algorithms in the table in real–time. The visual data comparison is shown in Figure 19.

## 6. Conclusions

This paper proposes a visual SLAM algorithm for complex dynamic environment. The lightweight convolutional neural network YOLOv5s is applied as the system parallel thread, and the lightweight network Ghostnet is applied as the backbone network of the target detection network YOLOv5s. Simultaneously, it is integrated into the attention mechanism module, which not only decreases the amount of model parameters and computing power requirements, improves the reasoning speed on the CPU, but also can better meet the challenges of complex dynamic environment. In the dynamic feature point removal section, an improved target detection network and logic discrimination module are combined to quickly remove the dynamic feature points in the SLAM fore–end, in an effort to improve the accuracy and efficiency of the entire SLAM system. This design balances the competitive demand between positioning accuracy and computational complexity. The experimental results on the TUM dataset show that contrast to the ORB–SLAM3 system, the pose estimation accuracy of our system has improved by 84.04%. Contrast with typical dynamic SLAM systems such as DS–SLAM, DVO SLAM, and DynaSLAM, our system has improved real–time property and accuracy. Although the current system performance has improved in terms of accuracy and real–time property, there is still massive work that needs to be further studied in the future. We plan to improve in two aspects: (1) further optimize the network model, while strengthening model training, cut and distill the network model to make the network more efficient and lightweight, and transplant the network to low–performing mobile devices, make it suitable for higher–level robot positioning work; (2) since the sparse point cloud atlas built by the system cannot be directly applied for navigation, we will complete the construction of octree map at the back–end of the system, further reflecting the high practicability and robustness of the system.

## Figures and Tables

**Figure 1 sensors-23-09274-f001:**
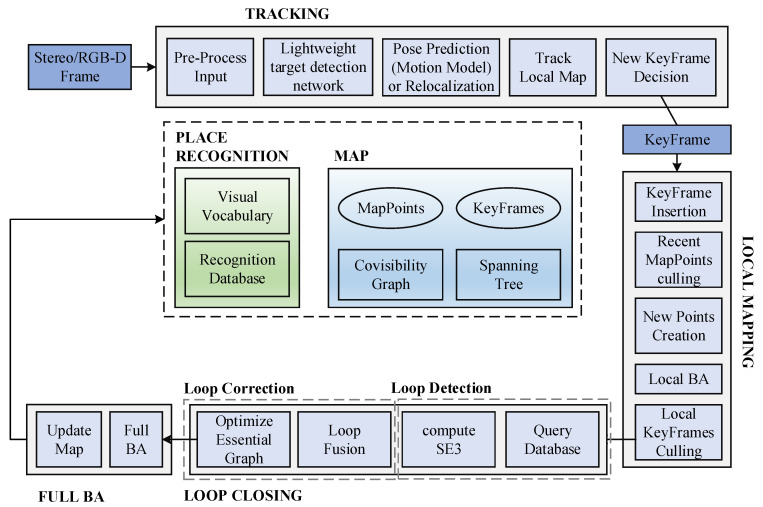
SLAM System Threads and Modules.

**Figure 2 sensors-23-09274-f002:**
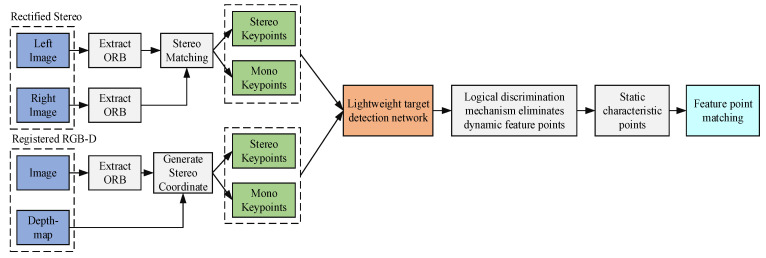
Image Preprocessing Process in the Tracking Thread.

**Figure 3 sensors-23-09274-f003:**
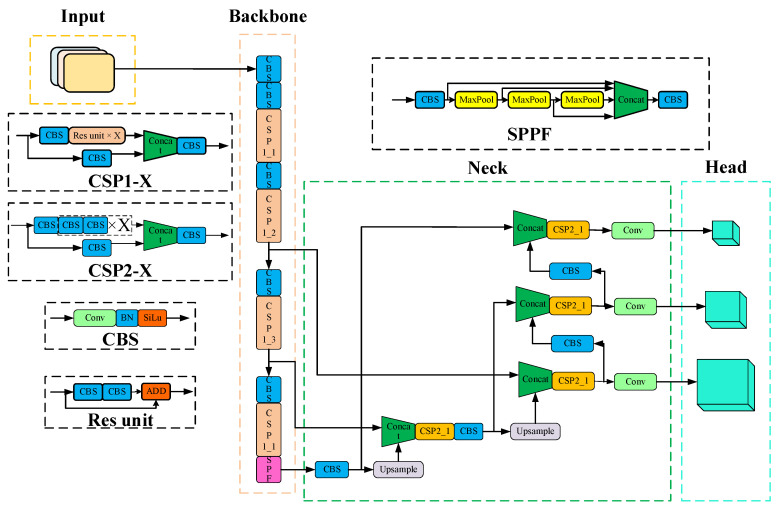
Original YOLOv5s network structure.

**Figure 4 sensors-23-09274-f004:**
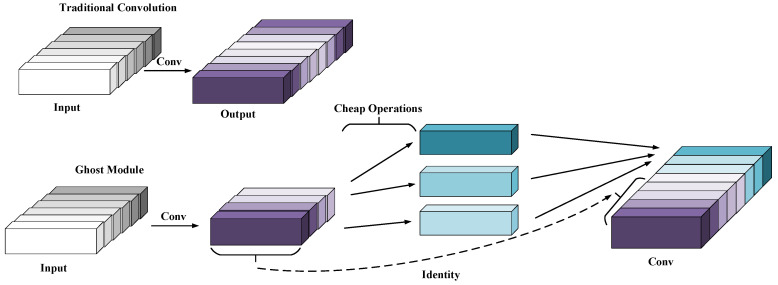
Comparison of Traditional convolution and Ghost convolution.

**Figure 5 sensors-23-09274-f005:**
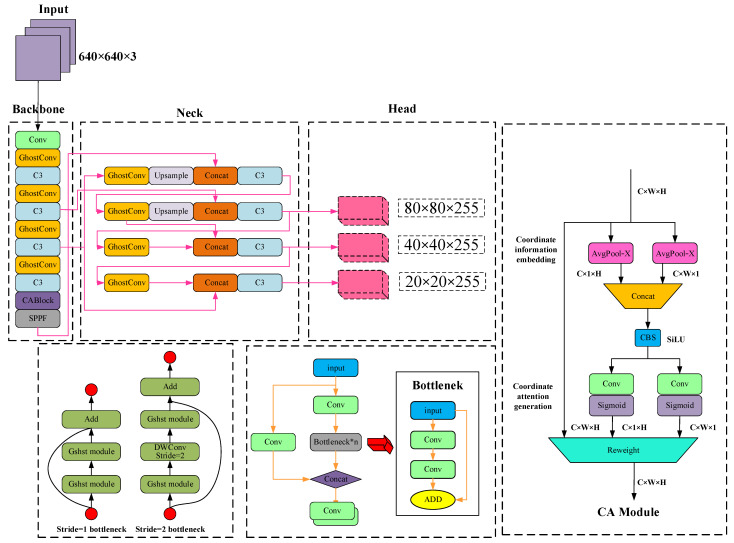
Overall structure of the proposed Lightweight Target Detection Network.

**Figure 6 sensors-23-09274-f006:**
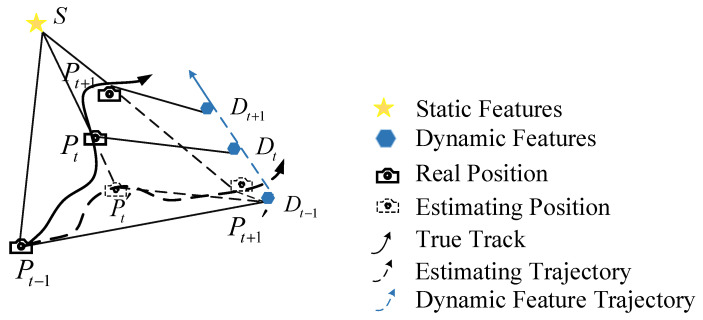
Impact of Dynamic Objects on the Pose Estimation of Visual Odometer in the scene.

**Figure 7 sensors-23-09274-f007:**
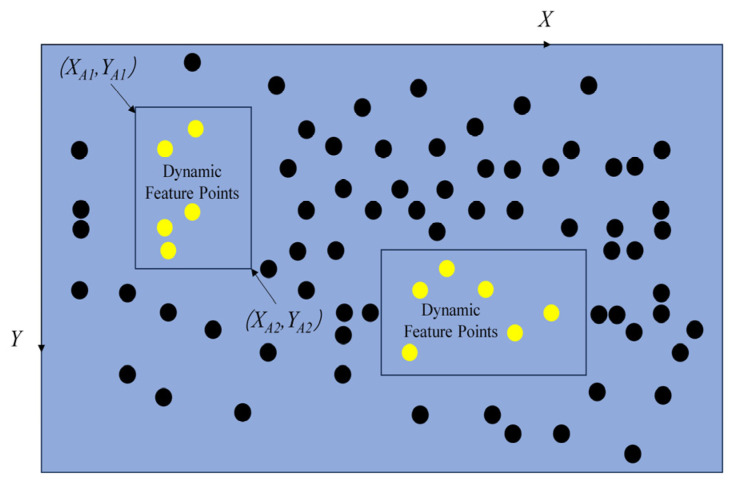
Distribution Schematic Diagram of Feature Points.

**Figure 8 sensors-23-09274-f008:**
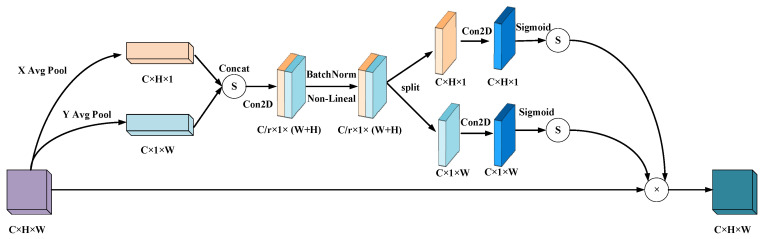
Structure of the coordinate attention mechanism.

**Figure 9 sensors-23-09274-f009:**
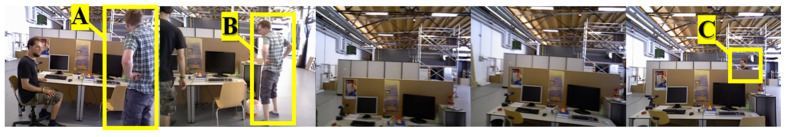
Changes in the Spatial Position of Dynamic Factors in Space.

**Figure 10 sensors-23-09274-f010:**
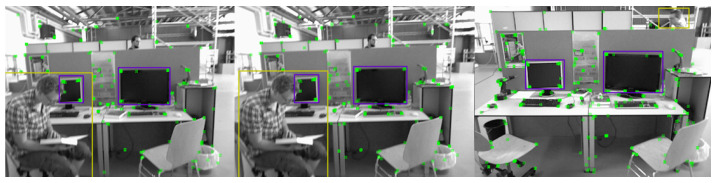
High Dynamic Objects Gain Higher Attention.(When a highly dynamic object is in a static position in space and moves further away from the camera, it can still be better captured by a lightweight target detection network) High dynamic objects are in the yellow box and low dynamic objects are in the purple box.

**Figure 11 sensors-23-09274-f011:**
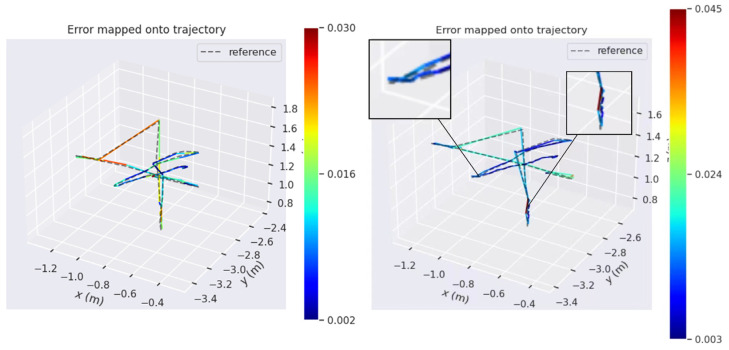
Comparison of Estimated Trajectories between Two Systems in the Walking_xyz Dataset (with attention mechanism module added on the left).

**Figure 13 sensors-23-09274-f013:**
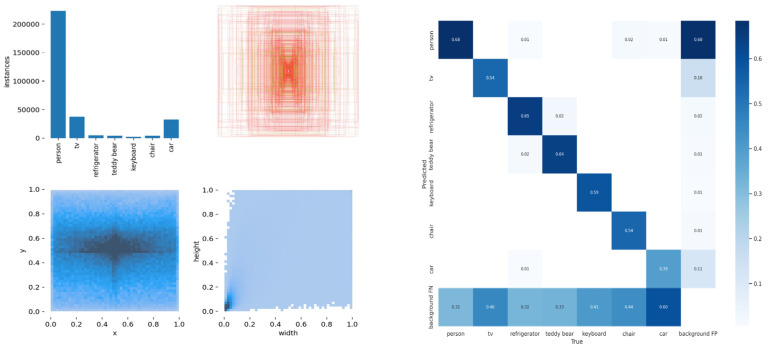
Training Sample Label and Confusion Matrix.

**Figure 14 sensors-23-09274-f014:**
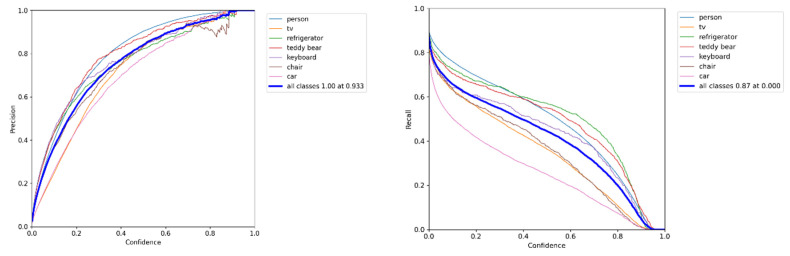
The diagram on the left shows the model accuracy and confidence, and the diagram on the right shows the recall rate and confidence.

**Figure 15 sensors-23-09274-f015:**
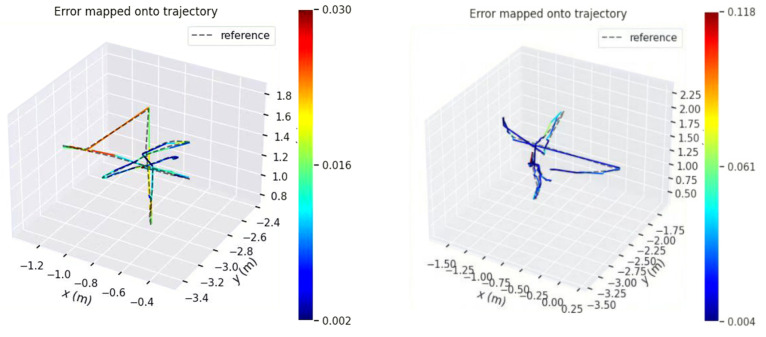
Comparison of Estimated Trajectories and Real Trajectories of the System on Walking_xyz and Walking_halfsphere Datasets in this paper.

**Figure 16 sensors-23-09274-f016:**
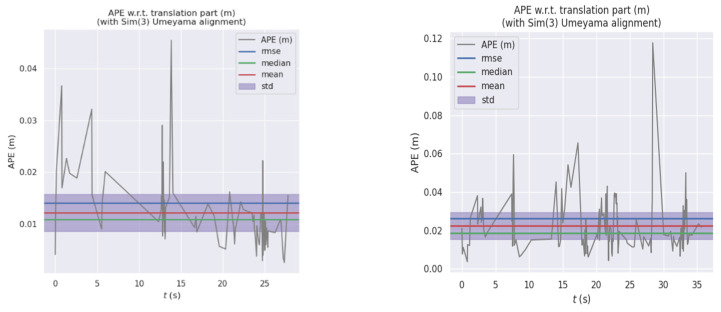
Error Distribution of the System in this paper on Walking_xyz and Walking_halfsphere Datasets.

**Figure 17 sensors-23-09274-f017:**
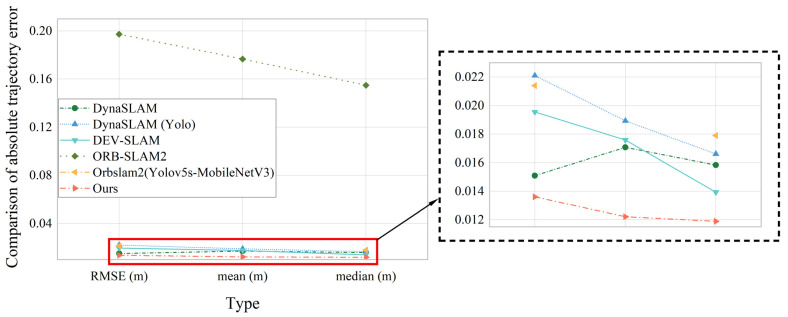
Comparition of AbsoluteTrajectory Error of Different Systems at Present.

**Figure 18 sensors-23-09274-f018:**
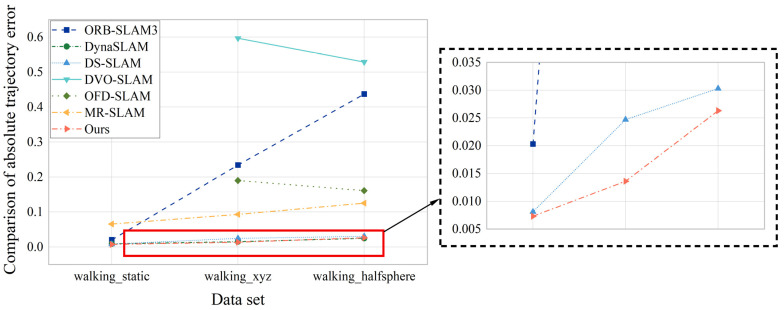
Comparison of RMSE Errors under Three Data for Different Systems.

**Figure 19 sensors-23-09274-f019:**
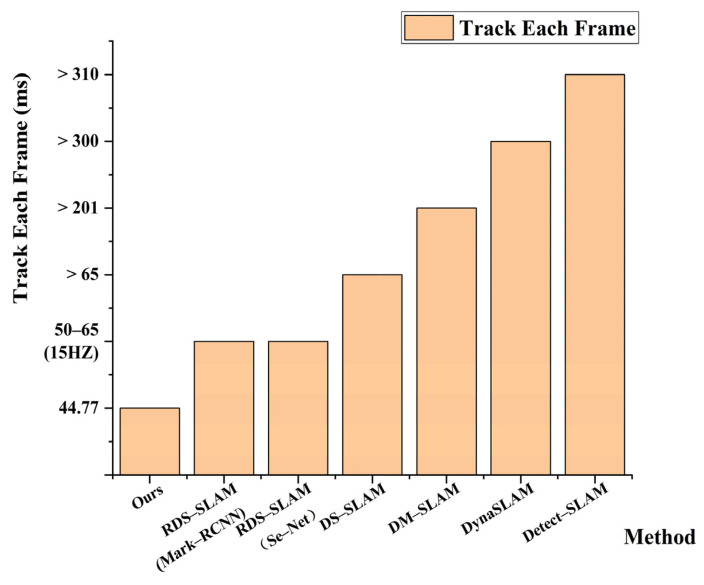
Comparison of average time spent per frame on tracking threads for different VSLAM systems.

**Table 1 sensors-23-09274-t001:** Mean and RMSE of ATE for RGB–D Cameras.

Type	RMSE (m)	Mean (m)	Median (m)	Dataset
DynaSLAM	0.01509	0.01707	0.01583	Walking_xyz
DynaSLAM (Yolo)	0.02210	0.01893	0.01661	
DEV–SLAM	0.01955	0.01759	0.01393	
ORB–SLAM2	0.19732	0.17668	0.15489	
Orbslam2(Yolov5s–MobileNetV3)	0.0214	—	0.01790	
Ours	0.0136	0.01221	0.01189	

**Table 2 sensors-23-09274-t002:** Comparison of Absolute Trajectory Error between Different System.

Data Set	ORB–SLAM3	DynaSLAM	DS–SLAM	DVO–SLAM	OFD–SLAM	MR–SLAM	Ours
walking_static	0.0203	0.0090	0.0081	—	—	0.0656	0.0073
walking_xyz	0.2341	0.0150	0.0247	0.5966	0.1899	0.0932	0.0136
walking_halfsphere	0.4372	0.0250	0.0303	0.5287	0.1612	0.1252	0.0263

**Table 3 sensors-23-09274-t003:** Comparison of Absolute Trajectory Errors between ORB–SLAM3 and the System in this paper.

Data Set	ORB–SLAM3				Ours				Relative Boost Ratio(%)			
	MEAN	MEDIAN	RMSE	STD	MEAN	MEDIAN	RMSE	STD	MEAN	MEDIAN	RMSE	STD
Walking_static	0.0169	0.0139	0.0203	0.0114	0.006579	0.006419	0.007311	0.003190	61.0710	53.8201	63.9852	72.0175
Walking_xyz	0.1989	0.1628	0.2341	0.1207	0.012211	0.011890	0.013665	0.006134	93.8607	92.6965	94.1627	94.9179
Walking_halfsphere	0.3662	0.2703	0.4372	0.2388	0.022386	0.018646	0.026355	0.013909	93.8869	93.1017	93.9718	94.1754

**Table 4 sensors-23-09274-t004:** Experiment results on GPU.

Network Model	Number of Parameters (M)	Computing Power (GELOPS)	Inference Time (S)	Frame Rate (FPS)
YOLOv5s–ghostnet–ca	3.34	6.2	21.28	38.80
YOLOv5s	7.46	17.5	24.04	34.12
YOLOv5m	21.79	52.3	26.46	31.25
YOLOv51	47.79	117.2	28.79	28.72
YOLOv5x	88.92	221.5	31.62	26.15

**Table 5 sensors-23-09274-t005:** Experiment results on CPU.

Network Model	Number of Parameters (M)	Computing Power (GELOPS)	Inference Time (S)	Frame Rate (FPS)
YOLOv5s–ghostnet–ca	3.34	6.2	29.91	27.65
YOLOv5s	7.46	17.5	34.92	23.68
YOLOv5m	21.79	52.3	57.71	14.33
YOLOv51	47.79	117.2	90.74	11.69
YOLOv5x	88.92	221.5	143.98	5.74

**Table 6 sensors-23-09274-t006:** The execution time under different deep learning–based algorithms.

Method	Semantic	Segmentation/Detection Time (ms)	Time of Other Models Related to Tracking (ms)	Track Each Frame (ms)
Detect–SLAM	SSD	310	Propagation: 20 Updating: 10	>310
DS–SLAM	SegNet	37.57330	ORB feature extraction: 9.375046 Moving consistency check: 29.50869	>65
DynaSLAM	Mask R–CNN	195	Multi–view Geometry: 235.98 Background in painting: 183.56	>300
DM–SLAM	Mask R–CNN	201.02	Ego–motion: 3.16 Dynamic Point Detection: 40.64	>201
RDS–SLAM	Mask R–CNN	200	Mask Generation: 5.42 Update Moving Probability: 0.17 Semantic–based Optimization: 0.54	50–65 (15 HZ)
RDS–SLAM	SegNet	30	Mask Generation: 5.04 Update Moving Probability: 0.17 Semantic–based Optimization: 0.50	50–65 (15 HZ)
Ours	YOLOv5s–ghostnet–ca	25.77	ORB tracking processing: 19	44.77

(Not every frame can be selected as a keyframe, so it is useful to keep track of how long it takes the thread to process each frame).

## Data Availability

Data are contained within the article.

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
