# Peer review of "A Lightweight Visual Simultaneous Localization and Mapping Method with a High Precision in Dynamic Scenes"

_sensors, 2023, doi:10.3390/s23229274_

Round 1

Reviewer 1 Report

Comments and Suggestions for Authors

Dear Authors,

thank you for your research.  I found few attention:

1. In line 38 please write more, expand the acronym IMU.

2. In line 45 is to much pouse.

3. Please think about Introduction because 16% write AI...please some change writing.

4. Please check in line 403 "is shown" please change.

5. In line 506 please check because you write about t but there isn't is in eq (10) in line 505.... I thing you must change symbol.

6. Please check in table 3 is Chines symbol, it is will be change.

7. Please check reference because there is many different symbols...

Thank you for attention

Reviewer 2 Report

Comments and Suggestions for Authors

The paper is proposing a new architecture for Visual Simultaneous Location And Mapping (VSLAM) which uses the ORB system to identify dynamic objects in the environment and thus improve the calculation of position in the environment from visual images.

I think that this is a solid paper but I have a few concerns that I think should be resolved in the final draft:

1.       The paper uses several abbreviations without defining them.  For example, while SLAM is defined, the term VSLAM is never defined in the paper.  The terms BA, CBS, ORB, R-CNN, and others are similarly undefined.  This makes the paper okay for those who are as involved in this research as the authors but makes the paper inaccessible to others.

2.       Figure 6 seems to have the variables Pt-1  , Pt, and Pt+1 assigned to two points each. I am assuming that one point is the true value and the other is the estimated value but I would recommend making this clearer.

3.       How exactly does Figure 11 show that the CA attention mechanism work better?  It seems that the error is the same for both graphs?  Am I missing something about the color being applied on the lines?  Some explanation is necessary.

4.       The paper needs to provide a better description of the split between training and evaluation sets used for the algorithm.  What are the relative sizes of the two sets?  What is the level of independence between the two sets?

5.       More information should be provided on the experiments which are used to evaluate the performance of the algorithm.  For example, what are the variety of scenarios for which tracking and mapping were performed?  How many tracking experiments were performed and what was the length of each tracking experiment? It might be nice to seem some plotting of the figures of merit versus motion speed for each of the algorithms to show how well the tracking of the proposed algorithm is improved over different motion types.

Again, I think that this is a good paper with only a few corrections/updates needed.

Comments on the Quality of English Language

There are only a few minor English issues. I had no troubles with the language use in this paper.

Reviewer 3 Report

Comments and Suggestions for Authors

There is too much work on the paper topic so it is essential to highlight novelties and contributions.

Abstract is very long so it should be shorthened and focus on essential issues.

Indroduction includes several references. but, aflter that section, there is another section with related work. I suggest to restructure these sections, shortehning the first one, focusing on the scope and challenges and include most of the references in the second one.

Algorithm explanation is fine.Perhaps, some parts included afterwards could be integrated in the explanation of the algorithm (for example, error measure)

Figure 3 and 5 are hard to read.

Relevant information in figure 11 is not clear. Some more explanations are requiered. Is it necesary to place this figure here when figure 15 appears in results section?

Figure 12 could be replaced by a table.

Revise Chineese characters in Table 3. Caomparisons are valuable.

Round 2

Reviewer 3 Report

Comments and Suggestions for Authors

MY comments have been considered in the revised version of the paper